# A Massive Scale Semantic Similarity Dataset of Historical English

**Emily Silcock**[1]**, Abhishek Arora**[1]**, Melissa Dell**[1,2][*]
[1]Harvard University; Cambridge, MA, USA.
[2]National Bureau of Economic Research; Cambridge, MA, USA.
[*]Corresponding author: melissadell@fas.harvard.edu.

## Abstract

A diversity of tasks use language models trained on semantic similarity data. While there are a variety of datasets that capture semantic similarity, they are either constructed from modern web data or are relatively small datasets created in the past decade by human annotators. This study utilizes a novel source, newly digitized articles from off-copyright, local U.S. newspapers, to assemble a massive-scale semantic similarity dataset spanning 70 years from 1920 to 1989 and containing nearly 400M positive semantic similarity pairs. Historically, around half of articles in U.S. local newspapers came from newswires like the Associated Press. While local papers reproduced articles from the newswire, they wrote their own headlines, which form abstractive summaries of the associated articles. We associate articles and their headlines by exploiting document layouts and language understanding. We then use deep neural methods to detect which articles are from the same underlying source, in the presence of substantial noise and abridgement. The headlines of reproduced articles form positive semantic similarity pairs. The resulting publicly available HEADLINES dataset is significantly larger than most existing semantic similarity datasets and covers a much longer span of time. It will facilitate the application of contrastively trained semantic similarity models to a variety of tasks, including the study of semantic change across space and time.

## 1 Introduction

Transformer language models contrastively trained on large-scale semantic similarity datasets are integral to a variety of applications in natural language processing (NLP). Contrastive training is often motivated by the anisotropic geometry of pre-trained transformer models like BERT [5], which complicates working with their hidden representations. Representations of low frequency words are pushed outwards on the hypersphere, the sparsity of low frequency words violates convexity, and the distance between embeddings is correlated with lexical similarity. This leads to poor alignment between semantically similar texts and poor performance when individual term representations are pooled to create a representation for longer texts [21]. Contrastive training reduces anisotropy [25].

A variety of semantic similarity datasets have been used for contrastive training [18]. Many of these datasets are relatively small, and the bulk of the larger datasets are created from recent web texts; *e.g.*, positive pairs are drawn from the texts in an online comment thread or from questions marked as duplicates in a forum. To provide a semantic similarity dataset that spans a much longer length of time and a vast diversity of topics, this study develops HEADLINES (**H**istorical **E**normous-Scale **A**bstractive **D**up**LI**cate **Ne**ws **S**ummaries), a massive dataset containing nearly 400 million high quality semantic similarity pairs drawn from 70 years of off copyright U.S. newspapers. Historically, around half of content in the many thousands of local newspapers across the U.S. was taken from centralized sources such as the Associated Press wire [8]. Local newspapers reprinted wire articles

37th Conference on Neural Information Processing Systems (NeurIPS 2023) Track on Datasets and Benchmarks.

but wrote their own headlines, which form abstractive summaries of the articles. Headlines written by different papers to describe the same wire article form positive semantic similarity pairs.

To construct HEADLINES, we digitize front pages of off-copyright local newspapers, localizing and OCRing individual content regions like headlines and articles. The headlines, bylines, and article texts that form full articles span multiple bounding boxes - often arranged with complex layouts - and we associate them using a model that combines layout information and language understanding [14]. Then, we use neural methods from [23] to accurately predict which articles come from the same underlying source, in the presence of noise and abridgement. HEADLINES allows us to leverage the collective writings of many thousands of local editors across the U.S., spanning much of the 20th century, to create a massive, high-quality semantic similarity dataset. HEADLINES captures semantic similarity with minimal noise, as positive pairs summarize the same underlying texts.

This study is organized as follows. Section 2 describes HEADLINES, and Section 3 relates it to existing datasets. Section 4 describes and evaluates the methods used for dataset construction, Section 5 benchmarks the dataset, and Section 6 discusses limitations and intended usage.

## 2 Dataset Description

HEADLINES contains 393,635,650 positive headline pairs from off-copyright newspapers. Figure 1 plots the distribution of content by state.

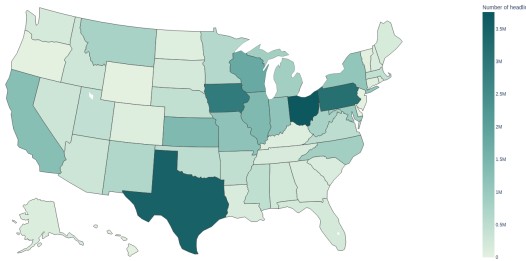

Figure 1: Geographic variation in source of headlines

Dataset statistics by decade are provided in Table 1. Content declines sharply in the late 1970s, due to a major copyright law change effective January 1, 1978.

| Decade | Headline Count | Cluster Count | Positive Pair Count | Word Count | Words Per Headline | Line Count | Lines Per Headline | Character Error Rate |
|---|---|---|---|---|---|---|---|---|
| 1920s | 4,889,942 | 1,032,108 | 28,928,226 | 68,486,589 | 14.0 | 18,983,014 | 3.9 | 4.3% |
| 1930s | 5,519,472 | 1,126,566 | 37,529,084 | 75,210,423 | 13.6 | 21,905,153 | 4.0 | 3.7% |
| 1940s | 6,026,940 | 1,005,342 | 62,397,004 | 61,629,003 | 10.2 | 19,538,729 | 3.2 | 2.4% |
| 1950s | 7,530,810 | 1,192,858 | 100,527,238 | 61,127,313 | 8.1 | 20,823,786 | 2.8 | 2.3% |
| 1960s | 6,533,071 | 926,819 | 108,415,279 | 46,640,311 | 7.1 | 16,408,148 | 2.5 | 3.7% |
| 1970s | 3,664,201 | 585,782 | 52,981,097 | 24,472,831 | 6.7 | 7,829,510 | 2.1 | 3.2% |
| 1980s | 703,052 | 170,507 | 2,857,722 | 5,161,537 | 7.3 | 1,502,893 | 2.1 | 1.5% |
| **Total** | **34,867,488** | **6,039,982** | **393,635,650** | **342,728,007** | **9.8** | **106,991,233** | **3.1** | |

Table 1: Descriptive statistics of HEADLINES.

The supplementary materials summarize copyright law for works first published in the United States. The newspapers in HEADLINES are off-copyright because they were published without a copyright notice or did not renew their copyright, required formalities at the time. Far from being an oversight, it was rare historically to copyright news, outside the nation's most widely circulated papers. The headlines in our dataset were written by editors at these local papers, and hence are in the public domain and anyone can legally use or reference them without permission.

It is possible that a newspaper not itself under copyright could reproduce copywritten content from some third party - the most prevalent example of this is comics - but this does not pose a problem for HEADLINES, since the dataset is built around the locally written headlines that describe the same

wire articles. If we were to accidentally include a syndicated headline, it would be dropped by our post-processing, since we drop headline pairs within a Levenshtein edit distance threshold of each other. It is also worth noting that a detailed search of U.S. copyright catalogs by [19] did not turn up a single instance of a wire service copyrighting their articles. (Even if they had, however, it would not pose a problem for headlines, since they were written locally.)

Figure 2 shows examples of semantic similarity pairs.

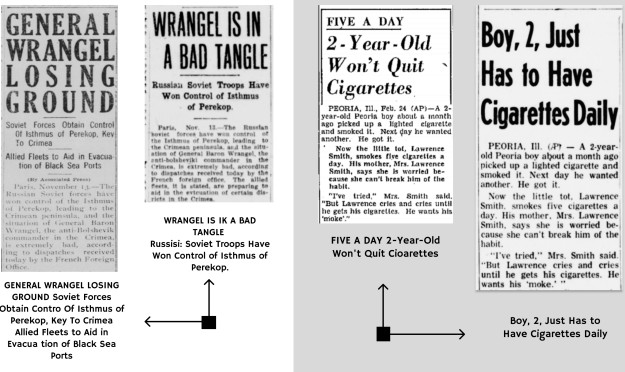

Figure 2: Semantic similarity examples, showing article image crops and OCR'ed headlines.

We quantify variation in HEADLINES across years, using a measure reminiscent of Earth Mover distance. This measure computes how much each text in a query dataset (e.g., 1920 headlines) would have to change (in embedding space) to have the same representation as the closest text in a key dataset (e.g., 1930 headlines).

Specifically, we first take a random sample of 10,000 texts per year. For year $j$, we embed texts $t_{1j}...t_{10,000j}$ using all-mpnet-base-v2. We choose MPNet because it has been shown to perform well across a variety of embedding datasets and tasks [18]. For each of these $t_{ij}$, we compute the most similar embedding in year $k$, measured by cosine similarity. This gives us a vector of similarity measures $s_{1jk}...s_{10,000jk}$, that for each text in year $j$ measure proximity to the most similar text in year $k$. We average these similarities to calculate $SIM_{jk}$.[1] Figure 3, which plots the $SIM_{jk}$, shows that similarity increases with temporal proximity. The dark square towards the upper left is World War 2, during which newspapers coverage was more homogeneous due to the centrality of the war.

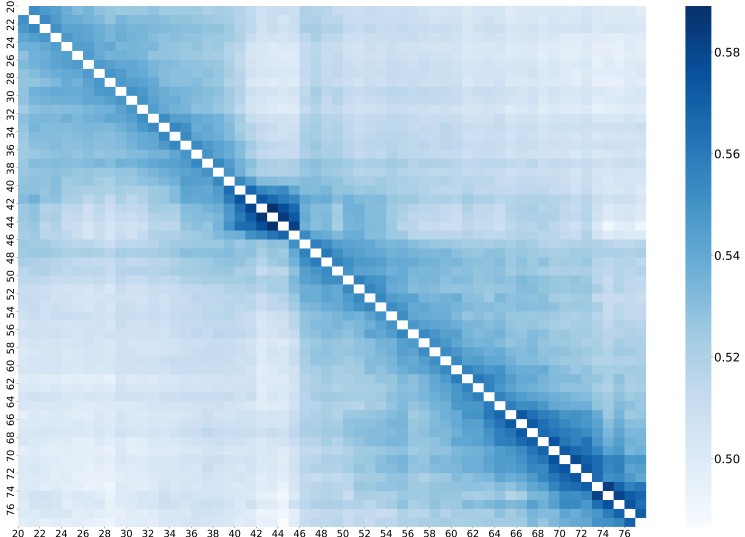

Figure 3: Average similarity between different years of HEADLINES.

[1]We end in 1977, as we have very few texts after this date.

`HEADLINES` is useful for training and evaluating models that aim to capture abstractive similarity, whether using the embeddings for tasks like clustering, nearest neighbor retrieval, or semantic search [18]. Because it contains chronological content over a long span of time, it can be used to evaluate dynamic language models for processing continuously evolving content [1, 15], as well as how large language models can be adapted to process historical content [17, 4, 16]. Likewise, it can be used to train or evaluate models that predict the region or year a text was written [20]. In addition, it is useful for training models and developing benchmarks for a number of downstream tasks, such as topic classification of vast historical and archival documents, which have traditionally been classified by hand. This is an extremely labor-intensive process, and as a result many historical archives and news collections remain largely unclassified. Similarly, it could facilitate creating a large scale dataset to measure term-level semantic change, complementing existing smaller-scale SemEval tasks.

`HEADLINES` has a Creative Commons CC-BY license, to encourage widespread use, and is available on Huggingface.[2]

## 3   Existing Semantic Similarity Datasets

There is a dense literature on semantic similarity, with datasets covering diverse types of textual similarity and varying greatly in size. The focus of `HEADLINES` on semantic similarity in historical texts sets it apart from other widely used datasets. It also dwarfs the size of most existing datasets, aggregating the collective work of 20th century newspapers editors, from towns across the U.S. Its paired headlines summarize the same text, rather than being related by other forms of similarity frequently captured by datasets, such as being in the same conversation thread or answering a corresponding question.

One related class of semantic similarity datasets consists of duplicate questions from web platforms, *e.g.*, questions tagged by users as duplicates from WikiAnswers (77.4 million positive pairs) [6], duplicate stack exchange questions (around 304,000 duplicate title pairs) [2], and duplicate quora questions (around 400,000 duplicate pairs) [12].[3] Alternatively, MS COCO [3] used Amazon's Mechanical Turk to collect five captions for each image in the dataset, resulting in around 828,000 positive caption pairs. In Flickr [26], 317,695 positive semantic similarity pairs describe around 32,000 underlying images. Like `HEADLINES`, positive pairs in these datasets refer to the same underlying content, but are describing an image rather than providing an abstractive summary of a longer text. In future work, `HEADLINES` could be expanded to include caption pairs describing the same underlying photo wire image, as local papers frequently wrote their own captions.

Online comment threads have also been used to train semantic similarity models. For example, the massive scale Reddit Comments [11] draws positive semantic similarity pairs from Reddit conversation threads between 2016 and 2018, providing 726.5 million positive pairs. Semantic similarity between comments in an online thread reflects conversational similarity, to the extent the thread stays on topic, rather than abstractive similarity. Likewise, question-answer and natural-language inference datasets are widely used for semantic similarity training.While other datasets exploit abstractive summaries - *e.g.*, Semantic Scholar (S2ORC) has been used to create semantic similarity pairs of the titles and abstracts of papers that cite each other - to our knowledge there are not large-scale datasets with abstractive summaries of the same underlying texts.

A wide variety of text embedding datasets have been combined into the Massive Text Embedding Benchmark (MTEB) [18], which evaluates 8 embedding tasks on 58 datasets covering 112 languages. We measure the similarity between `HEADLINES` and the English datasets in MTEB, using the Earth Mover-style distance, described in Section 2. As above, we first take a random sample of (up to) 10,000 texts from each decade of `HEADLINES`, as well as each of the English datasets in MTEB (if the dataset contains fewer than 10K texts, we use the full dataset and limit the comparison dataset to the same number of randomly selected texts). For dataset $j$, we embed texts $t_{1j}...t_{10,000j}$. For each of these $t_{ij}$, we compute the most similar embedding in dataset $k$, averaging these across all texts in $j$ to compute $SIM_{jk}$. $SIM_{jk}$ need not be symmetric. Suppose dataset $j$ is highly homogeneous, whereas dataset $k$ is heterogeneous. $SIM_{jk}$ may be high, because the similar embeddings in homogeneous dataset $j$ are close to a subset of embeddings in dataset $k$. On the other hand, $SIM_{kj}$ may be low, because most texts in dataset $k$ are dissimilar from texts in homogeneous dataset $j$.

---

[2]https://huggingface.co/datasets/dell-research-harvard/headlines-semantic-similarity
[3]Dataset sizes are drawn, when applicable, from a table documenting the training of Sentence BERT [21].

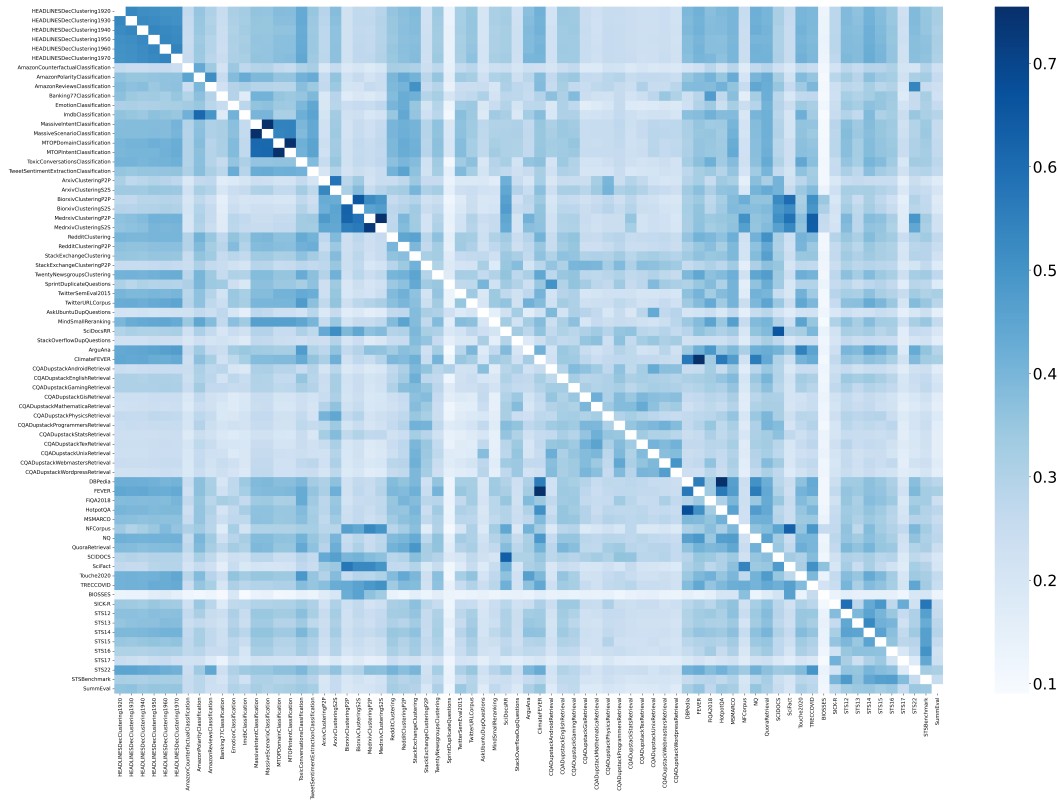

Figure 4: Average similarity between HEADLINES (by decade) and the English datasets in the Massive Text Embedding Benchmark. The similarity measure is described in the text.

Figure 4 shows the entire similarity matrix between HEADLINES and the English datasets in MTEB; rows are the query dataset and columns are the key. The style of the figure was adapted from [18]. The average similarity between HEADLINES and MTEB datasets is 31.4, whereas the average similarity between MTEB datasets and HEADLINES is 33.5, supporting our supposition that HEADLINES is diverse on average relative to other benchmarks. This average max similarity shows that there is ample information in HEADLINES not contained in existing datasets.

Table 2 shows examples where the nearest text to a headline in an MTEB dataset is highly similar, versus of average similarity. In some cases, highly similar texts in fact have a different meaning, but the limited context in the MTEB datasets makes this difficult to capture.

| Headline | Highly similar texts | Similarity |
|---|---|---|
| "Inflation Cuts are Questioned" | **Reddit**: "Today FOMC Resumed Meeting Inflation getting out of hand… Maybe… Maybe Not" | 0.55 |
| "Bear Bites Off Arm of Child" | **StackExchange**: "How to handle animal AI biting and holding onto the character... | 0.46 |
| "British Cruiser Reported Sunk" | **Twitter**: "That's It, Britain is Sunk" | 0.61 |
| "Will Free Press Dance to Government Tune" | **Twitter**: "Donald Trump v. a free press" | 0.60 |
| "Partitioning Plan Unsatisfactory, Ike Declares" | **Ubuntu Questions**: "Partitioning Issues" | 0.51 |
| Headline | Average similarity texts | Similarity |
| "Reds Knot Strong Tie" | **ArXiv**: "Knots and Polytopes" | 0.34 |
| "SHOWERS PROMISED TO END HEAT WAVE Two Deaths From Heat Over Weekend" | **StackOverflow**: "how to annotate heatmap with text in matplotlib?" | 0.27 |
| "Salary Boost Due For Some On Labor Day" | **Twitter**: "Glassdoor will now tell you if youre being underpaid" | 0.31 |
| "40 Old Ladies Now In Senate Says Rogers" | **Quora**: "Is 19 young?" | 0.28 |

Table 2: This table shows similarities between example texts.

# 4 Dataset Construction and Evaluation

## 4.1 Digitization

We digitized front pages from off-copyright newspapers spanning 1920-1989. We recognize layouts using Mask RCNN [10] and OCR the texts. We transcribed the headlines using Tesseract. The digitization was performed using Azure F-Series CPU nodes.

We evaluate this OCR on a hand annotated sample of 300 headlines per decade. Table 1 reports the character error rate, defined as the Levenshtein distance between the transcribed text and the ground truth, normalized by the length of the ground truth. As expected, OCR quality improves over time, as there is less damage from aging and fewer unusual fonts.

## 4.2 Article Association

Newspaper articles have complex and irregular layouts that can span multiple columns (Figure 5).

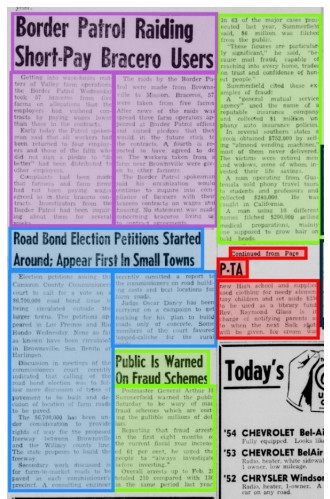 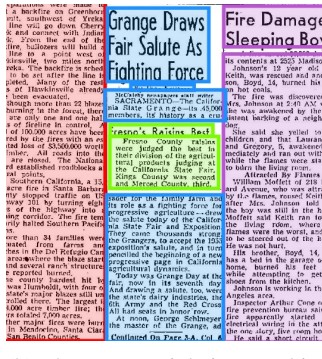 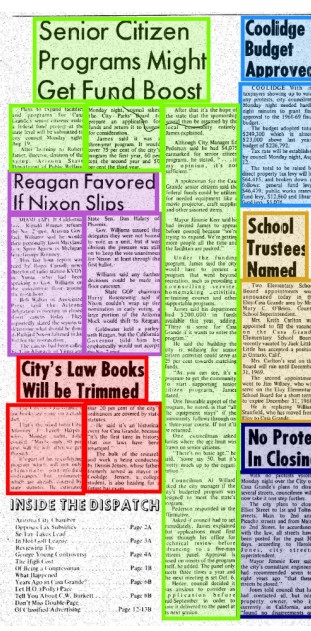

(a) The green article is not-contiguous and it is unclear if the second part continues the purple, green, or even blue articles if the text is not used.

(b) The green article is nested inside the blue article, which causes rule-based and image-based models to either associate the green and blue articles together, or miss the second half of the blue article.

(c) The smallest rectangle that contains the green article also contains the purple and red articles.

Figure 5: Articles that are misassociated with rule-based or image-based methods

We associate the (potentially multiple) headline bounding boxes with the (potentially multiple) article bounding boxes and byline boxes that comprise a single article using a combination of layout information and language understanding. A rule-based approach using the document layouts gets many of the associations correct, but misses some difficult cases where article bounding boxes are arranged in complex layouts. Language understanding can be used to associate such articles but must be robust to noise, from errors in layout detection (*e.g.* from cropping part of a content bounding box or adding part of the line below) and from OCR character recognition errors.

Hand annotating a sufficiently large training dataset would have been infeasibly costly. Instead, we devise a set of rules that - while recall is relatively low - have precision above 0.99, as measured on an evaluation set of 3,803 labeled bounding boxes. The algorithm exploits the positioning of article bounding boxes relative to headline boxes (as in Figure 6), first grouping an article bounding box with a headline bounding box if the rules are met and then associating all article bounding boxes grouped with the same headline together. Since precision is above 0.99, the rule generates nearly perfect silver-quality training data.

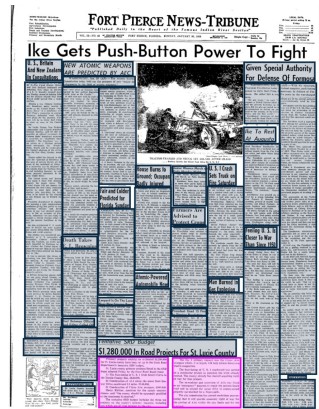

(a) Article bounding boxes that are under the same headline and at the bottom of the page are used to create training data.

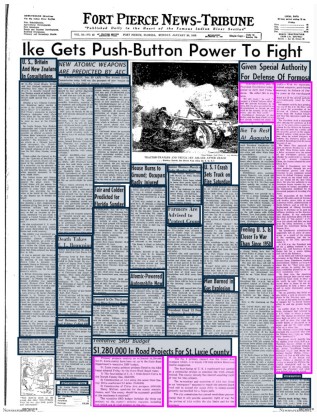

(b) At inference time, all bounding boxes that are directly under the same headline are associated with each other.

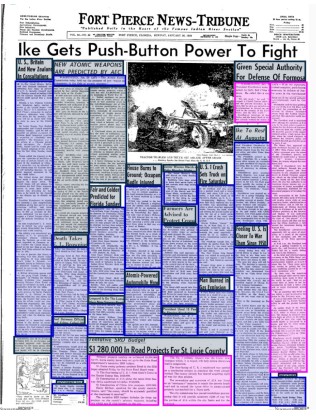

(c) Any other article with a headline directly above it is not compared, leaving only a few orphans that are left to be associated.

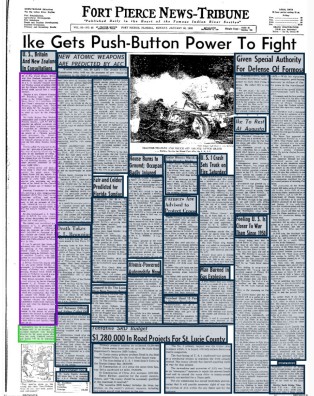

(d) This orphan (green bounding box) is compared with the bounding box above it. In this case, it is a separate article without a headline so it is not associated.

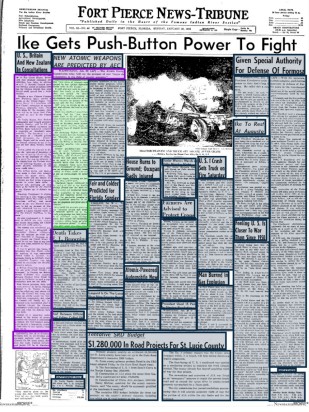

(e) Orphans are also compared with bounding boxes to the right. This orphan (green bounding box) is associated with the bounding box directly above.

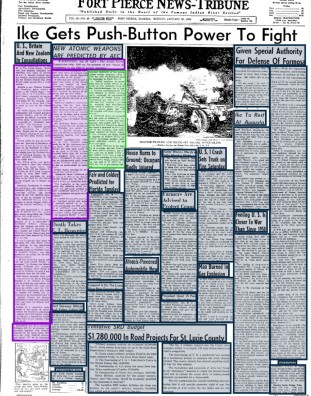

(f) The final orphan is compared with two columns to the right, as sometimes articles skip columns. If further columns to the right exist, they are not compared.

Figure 6: Illustration of article association pipeline

To train a language model to predict whether article box B follows box A, we embed the first and last 64 tokens of the texts in boxes B and A, respectively, with a RoBERTa base model [14]. The pair is positive when B follows A. The training set includes 12,769 positive associated pairs, with training details described in the supplementary materials. At inference time, we first associate texts using the rule-based approach, described in Figure 6, which has extremely high precision. To improve recall, we then apply the RoBERTa cross-encoder to remaining article boxes that could plausibly be associated, given their coordinates. Texts cannot be followed by a text that appears to the left, as layouts always proceed from left to right, so these combinations are not considered.

| | (1) F1 | (2) Recall | (3) Precision |
|---|---|---|---|
| Full Article Association | 93.7 | 88.3 | 99.7 |

Table 3: This table evaluates the full article association model.

We evaluate this method on a hand-labeled dataset of 214 scans. Full details of this dataset are given in the appendix. Table 3 evaluates recall, precision and F1 for associated articles. The F1 of 93.7

is high, and precision is extremely high. Errors typically occur when there is an error in the layout analysis or when contents are very similar, *e.g.*, grouping multiple obituaries into a single article.

## 4.3 Detecting Reproduced Content

Accurately detecting reproduced content can be challenging, as articles were often heavily abridged by local papers to fit within their space constraints and errors in OCR or article association can add significant noise. Table 4 shows examples of reproduced articles.

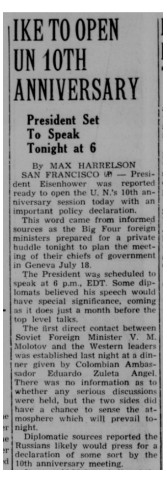

| | | | | |
|---|---|---|---|---|
| **IKE TO OPEN UN 10TH ANNIVERSARY** — President Set To Speak Tonight at 6 | SAN FRANCISCO (# — Presi-dent Eisenhower was_ reported ready to open the U. N.'s 10th an-niversary session today with an important policy declaration. This word came from in-formed sources as the Big Four foreign minis-ters prepared for a private huddle tonight to plan the meeting of their chiefs of government in Geneva July 18. The President was sched-uled to speak at 6 p.m., EDT. Some diplomats be-lieved his speech would have special significance, coming as it does just a month before the top level talks. The first direct contact between Soviet Foreign Minister V. M. Molotov and the Western lead-ers was established last night at a dinner given by Colombian Ambas-sador Eduardo Zuleta An-gel There was no infor-mation as tc whether any serious discussion: were held, but the two sides dic have a chance to sense the at-'}mosphere which will prevail to-'| night..| Diplomatic sources reported th« Russians likely would press for : 'Ideclaration of some sort by the | 10th anniversary meeting | **Big Four Ministers To Meet Tonight In San Francisco** | SAN FRANCISCO #—President Eisenhower was reported ready to open the U_N.'s 10th anniversary session today with an important policy declaration. This word came from in-formed sources as the Big Four foreign minis-ters prepared for a private huddle tonight to plan the meeting of their chiefs of government in Geneva July 18. Some diplomats believed Eisenhower's speech would have special significance, coming as it does just 'a month before the top-level talks.| The first direct contact between (Soviet Foreign Minister V. M.| Molotov and the Western leaders | was established last night "at a'dinner given by Colombian Am| bassador Eduardo Zulcta Angel. | Diplomatic sources reported the 'Russians likely would press for a declaration of some sort by the 110th anniversary meeting. | **Ike's Talk Will Open U. N. Meet** Big 4 Ministers Arrange Private Huddle Tonight | "SY MAA MARKRKRELSUN SAN FRANCISCO #®—President Eisen-hower was reported ready to open the U.N.'s 10th anniversary session today with an importan! policy declaration, This word came from in-formed sources as the Big Four foreign minis-ters prepared for a private huddle tonight to plan the meeting of their chiefs of government in Geneva July 18. Some diplomats believed Eisen.hower's speech would have special! significance, coming as it does just a month before the top-level talks. Mulles to Bo Host The first direct contact between Soviet Foreign Minister V. M. Molotov and the Western lead-ers was established last night at a dinner given by Colombian Ambassador Eduardo Zuleta Angel. Diplomatic sources reported the Russians likely would press for a 'declaration of some sort by the 10th anniversary meeting. |

Table 4: Examples of reproduced articles. Additions are highlighted, and OCR errors are underlined.

We use the model developed by [23], who show that a contrastively trained neural MPNet bi-encoder - combined with single linkage clustering of article representations - accurately and cheaply detects reproduced content. This bi-encoder is contrastively trained on a hand-labeled dataset (detailed in the appendix) to create similar representations of articles from the same wire source and dissimilar representations of articles from different underlying sources, using S-BERT's online contrastive loss [9] implementation.

We run clustering on the article embeddings by year over all years in our sample. In post-processing, we use a simple set of rules exploiting the dates of articles within clusters to remove content like weather forecasts and legal notices, that are highly formulaic and sometimes cluster together when they contain very similar content (*e.g.* a similar 5-day forecast) but did not actually come from the same underlying source. We remove all headline pairs that are below a Levenshtein edit distance, normalized by the min length in the pair, of 0.1, to remove pairs that are exact duplicates up to OCR noise. Training and inference were performed on an A6000 GPU card. More details are provided in the supplementary materials.

To evaluate how well the model detects reproduced content, we use a labeled sample of all front page articles appearing in the newspaper corpus for three days in the 1930s and 1970s, taken from [23]. This sample consists of 54,996 positive reproduced article pairs and 100,914,159 negative pairs. The

large-scale labeled evaluation dataset was generated using the above pipeline, so the evaluation is inclusive of any errors that result from upstream layout detection, OCR, or article association errors.

The neural bi-encoder methods achieve a high adjusted rand index (ARI) of 91.5, compared to 73.7 for an optimal local sensitive hashing specification, chosen on the validation set. This shows that our neural methods substantially outperform commonly used sparse methods for detecting reproduced content. The neural bi-encoder is slightly outperformed by adding a re-ranking step that uses a neural cross-encoder on the best bi-encoder matches (ARI of 93.7). We do not implement this method because the cross-encoder doesn't scale well. In contrast, the bi-encoder pipeline can be scaled to 10 million articles on a single GPU in a matter of hours, using a FAISS [13] backend.

|  | **Neural** | **Non-Neural** |
|---|---|---|
| **Most scalable** | Bi-encoder (91.5) | LSH (73.7) |
| **Less scalable** | Re-ranking (**93.7**) | $N$-gram overlap (75.0) |

Table 5: The numbers in parentheses are the Adjusted Rand Index for four different models - a bi-encoder, a "re-ranking" strategy that combines a bi- and cross-encoder, locally sensitive hashing (LSH), and $N$-gram overlap. Hyperparameters were chosen on the NEWS-COPY validation set, and all models were evaluated on the NEWS-COPY test set.

An error analysis is provided in [23]. Errors typically consist of articles about the same story from different wire services (*e.g.* the Associated Press and the United Press) or updates to a story as new events unfolded. Both types of errors will plausibly still lead to informative semantic similarity pairs.

## 5 Benchmarking

We benchmark HEADLINES using a variety of different language models and the MTEB clustering task. This task embeds texts using different base language models and then uses $k$ - the number of clusters in the ground truth data - for k-means clustering. Following MTEB, we score the model using the v-measure [22]. We should note that real-world problems are often framed as clustering tasks - rather than as classification tasks - because $k$ is unknown. By using $k$ from the ground truth, it makes the task easier. Nevertheless, we examine this task to allow for comparison with the rest of the literature.

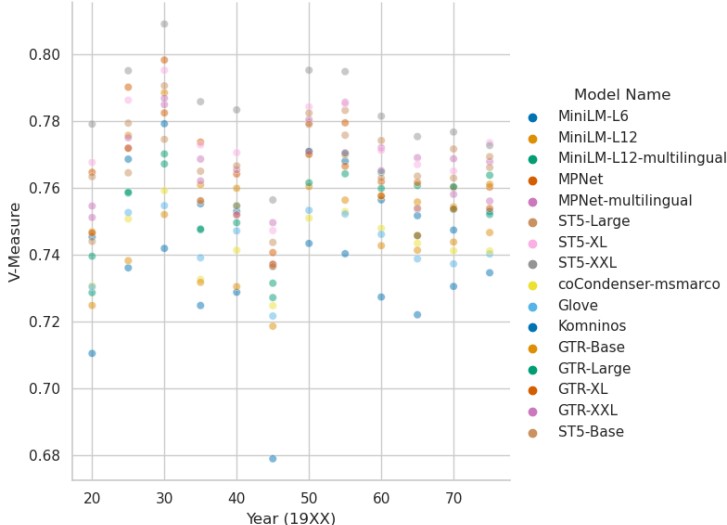

Figure 7: This figure benchmarks HEADLINES on the MTEB clustering task. The x-axis shows the year that the sample was taken from and the y-axis gives the v-measure.

Figure 7 plots the results of this benchmarking exercise. MTEB benchmarks clustering on Arxiv, Bioarxiv, Medarxiv, Reddit, StackExchange, and Twenty Newsgroups. Texts are labeled with their classification (e.g., fields like ComputerVision for the Arxiv datasets; the subreddit for Reddit). The

best average v-score across these datasets, from MPNet, is 43.69. The best average v-score across decades for HEADLINES, from ST5-XXL, is around 78. This difference is likely to reflect, at least in part, that our cluster labels are less noisy, since texts in the same cluster summarize the same content. In contrast, titles of Reddit posts in the same subreddit may be only loosely linked to each other, and many could be within the domain of another subreddit cluster. While a user happened to post in one subreddit, another user could have reasonably made the same titled post in a different subreddit. The under-identification of the clustering tasks for some texts in the MTEB datasets is suggested by the very low v-scores across state-of-the-art language models. Overall, this suggests the high quality of clusters in HEADLINES relative to many web text datasets. Yet there is still ample scope for improvement to the state-of-the-art model.

## 6    Limitations and Recommended Usage

HEADLINES contains some transcription errors. For working with historical texts, these are more a feature than a bug, as most historical texts are transcribed and also contain various OCR errors. Training a model on transcribed texts likely makes it more robust to transcription errors at inference time. However, researchers requiring completely clean texts should seek another corpus.

HEADLINES contains historical language, that reflects the semantics and cultural biases of many thousands of local newspaper editors. This is a distinguishing feature of HEADLINES, that is core to many potential applications. We do not attempt to filter texts with antiquated terms or that may be considered offensive, as this would invalidate the use of the dataset for studying semantic change and historical contexts. At the same time, this makes HEADLINES less suited for tasks that require texts that fully conform to current cultural standards or semantic norms. For these reasons, we recommend against the use of HEADLINES for training generative models. Rather, with nearly 400M positive semantic similarity pairs spanning much of the 20th century, it can plausibly play an important role in facilitating the application of large language models to historical texts.

## Acknowledgements

Funding was provided by the Harvard Data Science Initiative, Harvard Catalyst, and Microsoft Azure compute credits. We thank Luca D'Amico-Wong for excellent research assistance.

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

## Appendices

## A-1 Methods to Associate Articles

Figure 6 in the main text illustrates the full article association procedure.

First, we used a rule-based algorithm using associate article bounding boxes that are under the same headline, as these are part of the same article with extremely high probability. Algorithm 1 gives pseudocode for this method. We set the parameters as $P_S = 100$, $P_T = 20$, $P_B = 50$.

For training data, where we want article pairs that are not only part of the same article, but also where they appear in the given order, we further narrow down the pairs. Specifically, we use only those pairs which are horizontally next to each other, and which have no other bounding boxes below them, as for these pairs, we can guarantee that the pair of bounding follow directly after one another (whereas for other article bounding boxes that share a headline, there may be a third bounding box in between). Algorithm 2 shows pseudocode for this procedure, and we used $P_C = 5$, and it is further illustrated in panel A of figure 6 in the main text.

For hard negatives, we used article boxes under the same headline in reverse reading order (right to left). For standard negatives, we took pairs of articles on the same page, where B was above and to the left of A, as articles do not read from right to left. One twelfth of our training data were positive pairs, another twelfth were hard negative pairs and the remainder were standard negative pairs. This outperformed a more balanced training sample.

We use this dataset to finetune a cross-encoder using a RoBERTa base model [14]. We used a Bayesian search algorithm [7] to find optimal hyperparameters on one tenth of our training data (limited compute prevented us from running this search with the full dataset), which led to a learning rate of 1.7e-5, with a batch size of 64 and 29.57% warm up. We trained for 26 epochs with an AdamW optimizer, and optimize a binary cross-entropy loss.

We evaluate these methods on a hand-labeled dataset of 214 scans, randomly selected from 1968 and 1955. These scans were labeled by a highly-trained undergraduate research assistant. Summary statistics of this dataset are given in table A-1 and evaluation results are given in the main text.

| Scan count | Article bounding boxes | Headline bounding boxes | Article-article associations |
|---|---|---|---|
| 214 | 3,803 | 2,805 | 1,851 |

Table A-1: Descriptive statistics of article association training data.

## A-2 Methods to Detect Reproduced Content

To detect reproduced content, we use the contrastively trained bi-encoder model developed by [23], which is trained to learn similar representations for reproduced articles and dissimilar representations for non-reproduced articles. This model is based on an S-BERT MPNET model [21, 24] and is fine-tuned on a hand-labelled dataset of articles from the same underlying wire source, using S-BERT's online contrastive loss [9] implementation, with a 0.2 margin and cosine similarity as the distance metric. The learning rate is 2e-5 with 100% warm up and a batch size of 32. It uses an AdamW optimizer, and the model is trained for 16 epochs. This bi-encoder is trained and evaluated on a hand-labeled dataset, which is detailed in A-2. The results of this evaluation are given in the main text.

To create clusters from the bi-encoder embeddings, we use highly scalable single-linkage clustering, with a cosine similarity threshold of 0.94. We build a graph using articles as nodes, and add edges if the cosine similarity is above this threshold. As edge weights we use the negative exponential of the difference in dates (in days) between the two articles. We then apply Leiden community detection to the graph to control false positive edges that can otherwise merge disparate groups of articles.

We further remove clusters that have over 50 articles and contain articles with greater than five different dates. We also remove clusters that contain over 50 articles, when the number of articles is more than double the number of unique newspapers from which these articles are sourced. This

---

**Algorithm 1** Rule-based association of article bounding boxes

---

**INPUT:** $b_1, ..., b_n \in B$: set of bounding boxes that appear on the same scan, with their coordinates, denoted $left(b_i), right(b_i), top(b_i), bottom(b_i)$, and type (headline, article, byline etc.), denoted $type(b_i)$.

**PARAMETERS:**
    $W$: width of scan
    $H$: height of scan
    $P_S$: fraction of width, for creating side margin
    $P_T$: fraction of height, for creating top margin
    $P_B$: fraction of height, for creating bottom margin

**OUTPUT:** $ArticleArticlePairs = \{(b_i, b_j) \in B \times B | b_i, b_j$ predicted to be part of the same full article and $type(b_i) = type(b_j) = article\}$

1: Initialise: $M_S = W/P_S$, the side margin, $M_B = H/P_B$, the bottom margin, $M_T = H/P_T$, the top margin, $MatchedHeadlines = \{\}$, $HeadlineArticlePairs = \{\}$, $ArticleArticlePairs = \{\}$
2: **for** all $b_0$ in B where $type(b_0)$ is article **do**
3:     Create $B_0 \subset B$ where:
        - All bounding boxes of type byline are removed
        - $b_0$ is removed
        - All bounding boxes are removed that do not share at least $M_S$ of the horizontal axis
        - All bounding boxes are removed whose bottom is more than $M_B$ below the top of $b_0$
        - All bounding boxes are removed whose bottom is more than $M_T$ above the top of $b_0$
4:     **if** $B_0$ is not empty **then**
5:         Let $b_1$ be the element of $B_0$ that has the lowest bottom coordinate
6:         **if** $type(b_1)$ is headline **then**
7:             $MatchedHeadlines = MatchedHeadlines \cup \{b_1\}$
8:             $HeadlineArticlePairs = HeadlineArticlePairs \cup \{(b_0, b_1)\}$
9:         **end if**
10:     **end if**
11: **end for**
12: **for** all $b_h$ in $MatchedHeadlines$ **do**
13:     Let $H_1 \subset HeadlineArticlePairs$ be all pairs that contain that headline, $b_h$
14:     **if** $H_1$ has at least two elements **then**
15:         Let $A$ be all the bounding boxes of type article from the pairs in $H_1$
16:         Let $C$ be all combinations of 2 elements of $A$
17:         $ArticleArticlePairs = ArticleArticlePairs \cup C$
18:     **end if**
19: **end for**

---

---

**Algorithm 2** Selection of ordered article pairs

---

**INPUT:** $ArticleArticlePairs$, $B$, from algorithm 1.

**PARAMETERS:**
    $P_C$: fraction of column width, for creating margin

**OUTPUT:** $OrderedPairs \subset ArticleArticlePairs$

1: Initialise: $OrderedPairs = \{\}$
2: **for** $p$ in $ArticleArticlePairs$ **do**
3:     Let $p_l$ be the element of $p$ with the furthest left coordinate
4:     Let $p_r$ be the other element
5:     **if** $left(p_r)$ is not further to the right of $right(p_l)$ than $width(p_l)/F_C$ **then**
6:         **if** there are no other bounding boxes below $p_l$ **then**
7:             $OrderedPairs = OrderedPairs \cup \{p\}$
8:         **end if**
9:     **end if**
10: **end for**

---

removes clusters of content that are correctly clustered in the sense of being based on the same underlying source, but are not useful for the HEADLINES dataset. For example, an advertisement

| | Positives Pairs | Negative Pairs | Reproduced Articles | Singleton Articles | Total Articles |
|---|---|---|---|---|---|
| **Training Data** | | | | | |
| Training | 36,291 | 37,637 | 891 | – | 7,728 |
| Validation | 3,042 | 3,246 | 20 | – | 283 |
| **Full Day Evaluation** | | | | | |
| Validation | 28,547 | 12,409,031 | 447 | 2,162 | 4,988 |
| Test | 54,996 | 100,914,159 | 1,236 | 8,046 | 14,211 |
| **Full Dataset** | 122,876 | 113,364,073 | 2,594 | 10,208 | 27,210 |

Table A-2: Summary statistics of training and evaluation data for detecting duplicate content.

(misclassified as an article due to an article-like appearance) might be repeated by the same newspaper on multiple different dates and would be removed by these rules, or weather forecasts can be very near duplicates across space and time, forming large clusters.

## A-3 A Summary of Copyright Law for Works Published in the United States

| Date of Publication | Conditions | Copyright Term |
|---|---|---|
| ***Public Domain*** | | |
| Anytime | Works prepared by an officer/employee of the U.S. Government as part of their official duties | None |
| **Before 1928** | **None** | **None. Copyright expired.** |
| **1928 through 1977** | **Published without a copyright notice** | **None. Failure to comply with required formalities** |
| **1978 to 1 March 1989** | **Published without notice and without subsequent registration within 5 years** | **None. Failure to comply with required formalities** |
| **1928 through 1963** | **Published with notice but copyright was not renewed** | **None. Copyright expired** |
| ***Copyrighted*** | | |
| 1978 to 1 March 1989 | Published without notice, but with subsequent registration within 5 years | 70 (95) years after the death of author (corporate author) |
| 1928 through 1963 | Published with notice and the copyright was renewed | 95 years after publication |
| 1964 through 1977 | Published with notice | 95 years after publication |
| 1978 to 1 March 1989 | Created after 1977 and published with notice | 70 (95) years after the death of author (corporate author) or 120 years after creation, if earlier |
| 1978 to 1 March 1989 | Created before 1978 and first published with notice in the specified period | The greater of the term specified in the previous entry or 31 December 2047 |
| From 1 March 1989 through 2002 | Created after 1977 | 70 (95) years after the death of author (corporate author) or 120 years after creation, if earlier |
| From 1 March 1989 through 2002 | Created before 1978 and first published in this period | The greater of the term specified in the previous entry or 31 December 2047 |
| After 2002 | None | 70 (95) years after the death of author (corporate author) or 120 years after creation, if earlier |

Table A-3: This table summarizes U.S. copyright law, based on a similar table produced by the Cornell libraries. For concision, we focus on works initially published in the United States. A variety of other cases are also covered at `https://guides.library.cornell.edu/copyright`.

