# Supplementary Materials

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

This dataset has structured metadata following `schema.org`, and is readily discoverable.[1]

### S-4.2    DOI

The DOI for this dataset is: 10.57967/hf/0751.

### S-4.3    License

HEADLINES has a Creative Commons CC-BY license.

### S-4.4    Dataset usage

The dataset is hosted on huggingface, in json format. Each year in the dataset is divided into a distinct file (eg. 1952_headlines.json).

The data is presented in the form of clusters, rather than pairs to eliminate duplication of text data and minimize the storage size of the datasets.

An example from HEADLINES looks like:

```
{
    "headline": "FRENCH AND BRITISH BATTLESHIPS IN MEXICAN WATERS",
    "group_id": 4
    "date": "May-14-1920",
    "state": "kansas",
}
```

The data fields are:

- `headline`: headline text.
- `date`: the date of publication of the newspaper article, as a string in the form mmm-DD-YYYY.
- `state`: state of the newspaper that published the headline.
- `group_id`: a number that is shared with all other headlines for the same article. This number is unique across all year files.

The whole dataset can be easily downloaded using the `datasets` library:

```
from datasets import load_dataset
dataset_dict = load_dataset("dell-research-harvard/headlines-semantic-similarity")
```

Specific files can be downloaded by specifying them:

```
from datasets import load_dataset
load_dataset(
    "dell-research-harvard/headlines-semantic-similarity",
    data_files=["1929_headlines.json", "1989_headlines.json"]
)
```

---

[1]See `https://search.google.com/test/rich-results/result?id=_HKjxIv-LaF_8ElAarsM_g` for full metadata.

### S-4.5 Author statement

We bear all responsibility in case of violation of rights.

### S-4.6 Maintenance Plan

We have chosen to host HEADLINES on huggingface as this ensures long-term access and preservation of the dataset.

### S-4.7 Dataset documentation and intended uses

We follow the datasheets for datasets template [2].

#### S-4.7.1 Motivation

**For what purpose was the dataset created?** Was there a specific task in mind? Was there a specific gap that needed to be filled? Please provide a description.

*Transformer language models contrastively trained on large-scale semantic similarity datasets are integral to a variety of applications in natural language processing (NLP). A variety of semantic similarity datasets have been used for this purpose, with positive text pairs related to each other in some way. Many of these datasets are relatively small, and the bulk of the larger datasets are created from recent web texts; e.g. positives are drawn from the texts in an online comment thread or duplicate questions in a forum. Relative to existing datasets, HEADLINES is very large, covering a vast array of topics. This makes it useful generally speaking for semantic similarity pre-training. It also covers a long period of time, making it a rich training data source for the study of historical texts and semantic change. It captures semantic similarity directly, as the positive pairs summarize the same underlying texts.*

**Who created this dataset (e.g., which team, research group) and on behalf of which entity (e.g., company, institution, organization)?**
*HEADLINES was created by Melissa Dell and Emily Silcock, at Harvard University.*

**Who funded the creation of the dataset?** If there is an associated grant, please provide the name of the grantor and the grant name and number.

*The creation of the dataset was funded by the Harvard Data Science Initiative, Harvard Catalyst, and compute credits provided by Microsoft Azure to the Harvard Data Science Initiative.*

**Any other comments?**
*None.*

#### S-4.7.2 Composition

**What do the instances that comprise the dataset represent (e.g., documents, photos, people, countries)?** Are there multiple types of instances (e.g., movies, users, and ratings; people and interactions between them; nodes and edges)? Please provide a description.

*HEADLINES comprises instances of newspaper headlines and relationships between them. Specifically, each headline includes information on the text of the headline, the date of publication, and the state it was published in. Headlines have relationships between them if they are semantic similarity pairs, that is, if they two different headlines for the same newspaper article.*

**How many instances are there in total (of each type, if appropriate)?**
*HEADLINES contains 34,867,488 different headlines and 396,001,930 positive relationships between headlines.*

**Does the dataset contain all possible instances or is it a sample (not necessarily random) of instances from a larger set?** If the dataset is a sample, then what is the larger set? Is the sample representative of the larger set (e.g., geographic coverage)? If so, please describe how this representativeness was validated/verified. If it is not representative of the larger set, please describe why not (e.g., to cover a more diverse range of instances, because instances were withheld or unavailable).

*Many local newspapers were not preserved, and newspapers with the widest circulation tended to renew their copyrights, so cannot be included.*

**What data does each instance consist of? "Raw" data (e.g., unprocessed text or images) or features?** In either case, please provide a description.

*Each data instance consists of raw data. Specifically, an example from* `HEADLINES` *is:*

```
{
    "headline": "FRENCH AND BRITISH BATTLESHIPS IN MEXICAN WATERS",
    "group_id": 4
    "date": "May-14-1920",
    "state": "kansas",
}
```

*The data fields are:*

- `headline`*: headline text.*
- `date`*: the date of publication of the newspaper article, as a string in the form mmm-DD-YYYY.*
- `state`*: state of the newspaper that published the headline.*
- `group_id`*: a number that is shared with all other headlines for the same article. This number is unique across all year files.*

**Is there a label or target associated with each instance?** If so, please provide a description.

*Each instance contains a* `group_id` *as mentioned directly above. This is a number that is shared by all other instances that are positive semantic similarity pairs.*

**Is any information missing from individual instances?** If so, please provide a description, explaining why this information is missing (e.g., because it was unavailable). This does not include intentionally removed information, but might include, e.g., redacted text.

*In some cases, the state of publication is missing, due to incomplete metadata.*

**Are relationships between individual instances made explicit (e.g., users' movie ratings, social network links)?** If so, please describe how these relationships are made explicit.

*Relationships between instances are made explicit in the* `group_id` *variable, as detailed above.*

**Are there recommended data splits (e.g., training, development/validation, testing)?** If so, please provide a description of these splits, explaining the rationale behind them.

*There are no recommended splits.*

**Are there any errors, sources of noise, or redundancies in the dataset?** If so, please provide a description.

*The data is sourced from OCR'd text of historical newspapers. Therefore some of the headline texts contain OCR errors.*

**Is the dataset self-contained, or does it link to or otherwise rely on external resources (e.g., websites, tweets, other datasets)?** If it links to or relies on external resources, a) are there guarantees that they will exist, and remain constant, over time; b) are there official archival versions of the complete dataset (i.e., including the external resources as they existed at the time the dataset was created); c) are there any restrictions (e.g., licenses, fees) associated with any of the external resources that might apply to a future user? Please provide descriptions of all external resources and any restrictions associated with them, as well as links or other access points, as appropriate.

*The data is self-contained.*

**Does the dataset contain data that might be considered confidential (e.g., data that is protected by legal privilege or by doctor-patient confidentiality, data that includes the content of individuals non-public communications)?** If so, please provide a description.

*The dataset does not contain information that might be viewed as confidential.*

**Does the dataset contain data that, if viewed directly, might be offensive, insulting, threatening, or might otherwise cause anxiety?** If so, please describe why.

*The headlines in the dataset reflect diverse attitudes and values from the period in which they were written, 1920-1989, and contain content that may be considered offensive for a variety of reasons.*

**Does the dataset relate to people?** If not, you may skip the remaining questions in this section.

*Many news articles are about people.*

**Does the dataset identify any subpopulations (e.g., by age, gender)?** If so, please describe how these subpopulations are identified and provide a description of their respective distributions within the dataset.

*The dataset does not specifically identify any subpopulations.*

**Is it possible to identify individuals (i.e., one or more natural persons), either directly or indirectly (i.e., in combination with other data) from the dataset?** If so, please describe how.

*If an individual appeared in the news during this period, then headline text may contain their name, age, and information about their actions.*

**Does the dataset contain data that might be considered sensitive in any way (e.g., data that reveals racial or ethnic origins, sexual orientations, religious beliefs, political opinions or union memberships, or locations; financial or health data; biometric or genetic data; forms of government identification, such as social security numbers; criminal history)?** If so, please provide a description.

*All information that it contains is already publicly available in the newspapers used to create the headline pairs.*

**Any other comments?**
*None.*

### S-4.7.3 Collection Process

**How was the data associated with each instance acquired?** Was the data directly observable (e.g., raw text, movie ratings), reported by subjects (e.g., survey responses), or indirectly inferred/derived from other data (e.g., part-of-speech tags, model-based guesses for age or language)? If data was reported by subjects or indirectly inferred/derived from other data, was the data validated/verified? If so, please describe how.

*To create* `HEADLINES`, *we digitized front pages from off-copyright newspapers spanning 1920-1989. Historically, around half of articles in U.S. local newspapers came from newswires like the Associated Press. While local papers reproduced articles from the newswire, they wrote their own headlines, which form abstractive summaries of the associated articles. We associate articles and their headlines by exploiting document layouts and language understanding. We then use deep neural methods to detect which articles are from the same underlying source, in the presence of substantial noise and abridgement. The headlines of reproduced articles form positive semantic similarity pairs.*

**What mechanisms or procedures were used to collect the data (e.g., hardware apparatus or sensor, manual human curation, software program, software API)?** How were these mechanisms or procedures validated?

*These methods are described in detail in the main text and supplementary materials of this paper.*

**If the dataset is a sample from a larger set, what was the sampling strategy (e.g., deterministic, probabilistic with specific sampling probabilities)?**
*The dataset was not sampled from a larger set.*

**Who was involved in the data collection process (e.g., students, crowdworkers, contractors) and how were they compensated (e.g., how much were crowdworkers paid)?**
*We used student annotators to create the validation sets for associating bounding boxes, and the training and validation sets for clustering duplicated articles. They were paid $15 per hour, a rate set by a Harvard economics department program providing research assistantships for undergraduates.*

**Over what timeframe was the data collected? Does this timeframe match the creation timeframe of the data associated with the instances (e.g., recent crawl of old news articles)?** If not, please describe the timeframe in which the data associated with the instances was created.

*The headlines were written between 1920 and 1989. Semantic similarity pairs were computed in 2023.*

**Were any ethical review processes conducted (e.g., by an institutional review board)?** If so, please provide a description of these review processes, including the outcomes, as well as a link or other access point to any supporting documentation.

*No, this dataset uses entirely public information and hence does not fall under the domain of Harvard's institutional review board.*

**Does the dataset relate to people?** If not, you may skip the remaining questions in this section.

*Historical newspapers contain a variety of information about people.*

**Did you collect the data from the individuals in question directly, or obtain it via third parties or other sources (e.g., websites)?**
*The data were obtained from off-copyright historical newspapers.*

**Were the individuals in question notified about the data collection?** If so, please describe (or show with screenshots or other information) how notice was provided, and provide a link or other access point to, or otherwise reproduce, the exact language of the notification itself.

*Individuals were not notified; the data came from publicly available newspapers.*

**Did the individuals in question consent to the collection and use of their data?** If so, please describe (or show with screenshots or other information) how consent was requested and provided, and provide a link or other access point to, or otherwise reproduce, the exact language to which the individuals consented.

*The dataset was created from publicly available historical newspapers.*

**If consent was obtained, were the consenting individuals provided with a mechanism to revoke their consent in the future or for certain uses?** If so, please provide a description, as well as a link or other access point to the mechanism (if appropriate).

*Not applicable.*

**Has an analysis of the potential impact of the dataset and its use on data subjects (e.g., a data protection impact analysis) been conducted?** If so, please provide a description of this analysis, including the outcomes, as well as a link or other access point to any supporting documentation.

*No.*

**Any other comments?**
*None.*

### S-4.7.4 Preprocessing/cleaning/labeling

**Was any preprocessing/cleaning/labeling of the data done (e.g., discretization or bucketing, tokenization, part-of-speech tagging, SIFT feature extraction, removal of instances, processing of missing values)?** If so, please provide a description. If not, you may skip the remainder of the questions in this section.

*See the description in the main text.*

**Was the "raw" data saved in addition to the preprocessed/cleaned/labeled data (e.g., to support unanticipated future uses)?** If so, please provide a link or other access point to the "raw" data.

*No.*

**Is the software used to preprocess/clean/label the instances available?** If so, please provide a link or other access point.

*No specific software was used to clean the instances.*

**Any other comments?**
*None.*

### S-4.7.5 Uses

**Has the dataset been used for any tasks already?** If so, please provide a description.

*No.*

**Is there a repository that links to any or all papers or systems that use the dataset?** If so, please provide a link or other access point.

*No.*

**What (other) tasks could the dataset be used for?**
*The dataset can be used for training models for semantic similarity, studying language change over time and studying difference in language across space.*

**Is there anything about the composition of the dataset or the way it was collected and preprocessed/cleaned/labeled that might impact future uses?** For example, is there anything that a future user might need to know to avoid uses that could result in unfair treatment of individuals or groups (e.g., stereotyping, quality of service issues) or other undesirable harms (e.g., financial harms,

legal risks) If so, please provide a description. Is there anything a future user could do to mitigate these undesirable harms?

*The dataset contains historical news headlines, which will reflect current affairs and events of the time period in which they were created, 1920-1989, as well as the biases of this period.*

**Are there tasks for which the dataset should not be used?** If so, please provide a description.

*It is intended for training semantic similarity models and studying semantic variation across space and time.*

**Any other comments?**
*None*

### S-4.7.6  Distribution

**Will the dataset be distributed to third parties outside of the entity (e.g., company, institution, organization) on behalf of which the dataset was created?** If so, please provide a description.

*Yes. The dataset is available for public use.*

**How will the dataset will be distributed (e.g., tarball on website, API, GitHub)** Does the dataset have a digital object identifier (DOI)?

*The dataset is hosted on huggingface. Its DOI is 10.57967/hf/0751.*

**When will the dataset be distributed?**
*The dataset was distributed on 7th June 2023.*

**Will the dataset be distributed under a copyright or other intellectual property (IP) license, and/or under applicable terms of use (ToU)?** If so, please describe this license and/or ToU, and provide a link or other access point to, or otherwise reproduce, any relevant licensing terms or ToU, as well as any fees associated with these restrictions.

*The dataset is distributed under a Creative Commons CC-BY license. The terms of this license can be viewed at* `https://creativecommons.org/licenses/by/2.0/`

**Have any third parties imposed IP-based or other restrictions on the data associated with the instances?** If so, please describe these restrictions, and provide a link or other access point to, or otherwise reproduce, any relevant licensing terms, as well as any fees associated with these restrictions.

*There are no third party IP-based or other restrictions on the data.*

**Do any export controls or other regulatory restrictions apply to the dataset or to individual instances?** If so, please describe these restrictions, and provide a link or other access point to, or otherwise reproduce, any supporting documentation.

*No export controls or other regulatory restrictions apply to the dataset or to individual instances.*

**Any other comments?**
*None.*

### S-4.7.7  Maintenance

**Who will be supporting/hosting/maintaining the dataset?**

*The dataset is hosted on huggingface.*

**How can the owner/curator/manager of the dataset be contacted (e.g., email address)?**

*The recommended method of contact is using the huggingface 'community' capacity. Additionally, Melissa Dell can be contacted at melissadell@fas.harvard.edu.*

**Is there an erratum?**   If so, please provide a link or other access point.

*There is no erratum.*

**Will the dataset be updated (e.g., to correct labeling errors, add new instances, delete instances)?** If so, please describe how often, by whom, and how updates will be communicated to users (e.g., mailing list, GitHub)?

*We have no plans to update the dataset. If we do, we will notify users via the huggingface Dataset Card.*

**If the dataset relates to people, are there applicable limits on the retention of the data associated with the instances (e.g., were individuals in question told that their data would be retained for a fixed period of time and then deleted)?**   If so, please describe these limits and explain how they will be enforced.

*There are no applicable limits on the retention of data.*

**Will older versions of the dataset continue to be supported/hosted/maintained?**   If so, please describe how. If not, please describe how its obsolescence will be communicated to users.

*We have no plans to update the dataset. If we do, older versions of the dataset will not continue to be hosted. We will notify users via the huggingface Dataset Card.*

**If others want to extend/augment/build on/contribute to the dataset, is there a mechanism for them to do so?**   If so, please provide a description. Will these contributions be validated/verified? If so, please describe how. If not, why not? Is there a process for communicating/distributing these contributions to other users? If so, please provide a description.

*Others can contribute to the dataset using the huggingface 'community' capacity. This allows for anyone to ask questions, make comments and submit pull requests. We will validate these pull requests. A record of public contributions will be maintained on huggingface, allowing communication to other users.*

**Any other comments?**
*None.*