# OpenReview forum: "A Massive Scale Semantic Similarity Dataset of Historical English"
_NeurIPS.cc/2023/Track/Datasets_and_Benchmarks — NeurIPS 2023 Datasets and Benchmarks Poster_

### Official Review · Reviewer_oH6a · 2023-07-21
**large historical semantic similarity dataset in English**

**Rating:** 6
**Confidence:** 4
**Clarity:** Yes, indeed.

**Strengths:**

1- The paper is well-written and easy to follow.
2 - The dataset is innovative and large.
3 - The dataset is accessible and publicly available

**Additional Feedback:**

Here are a few questions:
1 - What makes historical local news different from "recent web texts"? if yes, please explain. Then, Can you quantify that?
2 - What is the effect of geographic variation in sources of headlines? Have you measured the effect?

**Correctness:**

Yes, the claim is valid and sound but there is no benchmark or through evaluation. Therefore, it is hard to use the dataset if those mentioned errors  in the paper are not quantified or evaluated.

**Documentation:**

The authors explain thoroughly how the data can be collected but there is not enough information for reproducibility.

**Ethics:**

There are no issues.

**Limitations:**

The authors have an have not addressed the limitations clearly. There is no defined section as "Limitations" . However, the authors recognizes that the OCR method can result into errors that cause grammatical and spelling issues which may affect the context of the article.

**Opportunities For Improvement:**

Although the paper presents a significant contribution to the field of NLP by digitizing historical local newspapers, the paper can be improved if the following issues were addressed:
1 - Benchmark the dataset. This has a significant negative impact on the paper.
2 - The paper could utilize a lot of space in the paper for writing more beneficial consent such as experiments and benchmarking. For example, Figure 1 and Figure 3 are too large and they take up too much space. This is just an example of many.
3 - The paper does not include tables or clear basic statistical facts of their dataset. For example, average work count, sentence count, etc. Those are just examples.
4 - The authors can show a pseudo code to describe the rule-based method they used.

**Relation To Prior Work:**

This section causes a weakness in the paper. The comparison drawn between their work and existing work does not draw a clear related comparison. For example, they compare their textual data to an image dataset. It would be acceptable if they authors present the similarity between them. Further, it would be great the paper includes a table that compares the existing datasets and this paper's dataset.

**Summary And Contributions:**

The paper provides HEADLINES (Historical Enormous-Scale Abstractive DupLIcate News Summaries). It is a large dataset encompassing nearly 400,000 semantic similarity pairs drawn from nearly 70 years of off copyright U.S. newspapers. The authors automated a method to extract U.S. local newspapers articles from 1920 to 1989. The author also used deep neural techniques to put the scattered pieces of the articles together.  The contribution of the paper is the following:

1 - The authors provide a large and accessible dataset of 400K semantic similarity pairs from a 70 years long period from U.S. local newspapers.
2 - They digitized the historical newspapers' articles.

---

> ### Author Response · Authors · 2023-08-22
> **Author Response**
>
> We thank the reviewer for their thoughtful comments. We direct all reviewers to our updated submission, and respond individually here to the points raised in this review. This comment is part 1 of our response.
>
> *Benchmark the dataset*
>
>  Response: We have now added benchmarking  (Section 5 and Figure 7), following the Massive Text Embedding Benchmark (MTEB), the primary benchmark in this domain. We use the MTEB clustering task, which embeds texts using different base language models and then uses $k$ - the number of clusters in the ground truth data - for k-means clustering. Following MTEB, we score the model using the v-measure (Rosenberg, 2007). In many real-world tasks, a problem is framed as a clustering task rather than as a classification task because $k$ is unknown. Nevertheless, we use this easier task where $k$ is assumed to be known for comparability to the literature.
>
> MTEB benchmarks clustering on Arxiv, Bioarxiv, Medarxiv, Reddit, StackExchange, and Twenty Newsgroups datasets. Texts are labeled with their classification (e.g., fields like ComputerVision for the Arxiv datasets; the subreddit for Reddit). The best average v-score across these datasets, from MPNet, is 43.69. The best average v-score across decades for HEADLINES, from ST5-XXL, is around 78 (Figure 7). This difference, while it could be interpreted as evidence that HEADLINES is easier, is also likely to reflect the fact that our cluster labels are less noisy, since texts in the same cluster summarize the same content. In contrast, titles of Reddit posts in the same subbredit may be only loosely (or not at all) linked to each other, and many could be within the domain of another subbredit cluster. While a user happened to post in one subreddit, another user could have reasonably made the same titled post in a different subreddit. The under-identification of the clustering tasks for some of the texts in MTEB is suggested by the very low v-scores across state-of-the-art language models. Overall, this suggests the high quality of HEADLINES relative to many web text datasets.  Yet there is still ample scope for improvement to the state-of-the-art model.
>
> *Dataset Statistics*
>
> Response: Table 1 now provides these statistics.
>
> *Pseudo code to describe the rule-based method*
>
> Response: This is now included in the supplementary material, as algorithms 1 and 2.
>
> “There is no defined section as "Limitations" . However, the authors recognizes that the OCR method can result into errors...”
>
> Response: We have added Section 6 on Limitations. Table 1 now includes an assessment of OCR errors, using 300 randomly selected, hand-annotated texts per decade. The character error rate ranges from 4.3% (1920s) to 1.5% (1980s), for the most part declining monotonically across time.  We do not think OCR errors are overly-detrimental to dataset quality, as suggested by the strong performance on the benchmarking exercise (Figure 7 and Section 5).

---

> ### Author Response · Authors · 2023-08-22
> **Author Response: Part 2**
>
> This is part 2 of our response:
>
> *“The comparison drawn between their work and existing work does not draw a clear related comparison. For example, they compare their textual data to an image dataset. It would be acceptable if they authors present the similarity between them. Further, it would be great the paper includes a table that compares the existing datasets and this paper's dataset.”*
>
> Response: We  measure the similarity between HEADLINES and the English datasets in MTEB, using a measure reminiscent of Earth Mover distance. We first take a random sample of (up to) 10,000 texts from each decade of HEADLINES, as well as each of the English datasets in MTEB (if the dataset contains fewer than 10K texts, we use the full dataset and limit the comparison dataset to the same number of randomly selected texts). For dataset $j$, we embed texts $t_{1j}...t_{10,000j}$. For each of these $t_{ij}$, we compute the most similar embedding in dataset $k$, averaging these across all texts in $j$ to compute $SIM_{jk}$.
> $SIM_{jk}$ need not be symmetric. Suppose dataset $j$ is highly homogeneous, whereas dataset $k$ is heterogeneous. $SIM_{jk}$ may be high, because the similar embeddings in homogeneous dataset $j$ are close to a subset of embeddings in dataset $k$. On the other hand, $SIM_{kj}$ may be low, because most texts in dataset $k$ are dissimilar from texts in homogeneous dataset $j$.
>
> Figure 4 shows the similarity matrix between HEADLINES and the English datasets in MTEB; rows are the query dataset and columns are the key. The style of the figure was adapted from MTEB.  The average similarity between HEADLINES and MTEB datasets is 31.4, whereas the average similarity between MTEB datasets and HEADLINES is 33.5, supporting our supposition that HEADLINES is diverse relative to other benchmarks. This average max similarity shows that there is ample information in HEADLINES not contained in existing datasets.
>
> Table 2 shows examples where the nearest text to a headline in an MTEB dataset is highly similar, versus of average similarity. In some cases, highly similar texts in fact have a different meaning, but the limited context in the MTEB datasets makes this difficult to capture.
>
> Figure 3 conducts a similar exercise across years of HEADLINES, showing that similarity declines with the length of time that has elapsed between texts.
>
> *“The authors explain thoroughly how the data can be collected but there is not enough information for reproducibility.”*
>
> Response: The supplementary materials have been expanded and now contain all information required to reproduce the results.
>
> *“What makes historical local news different from "recent web texts"?*
>
> Response: The exercise comparing HEADLINES to existing benchmarks aims to quantify the differences. Table 2 gives an intuition for these differences by providing example texts.
>
> *“What is the effect of geographic variation in sources of headlines? Have you measured the effect?”*
>
> Response: Quantifying geographic variation in semantics is a fascinating question. However, to our knowledge there are not existing off-the-shelf methods to compellingly measure this variation (the literature largely relies on static word embeddings). We hope that releasing HEADLINES will facilitate more research on this important question by providing a novel, massive scale historical dataset that identifies the geographic origin of the text.

---

### Official Review · Reviewer_eGCu · 2023-07-22
**Large dataset for semantic similarity task assessment**

**Rating:** 8
**Confidence:** 4

**Strengths:**

Large dataset with newspaper articles and headlines across a wide variety of topics.
Introduce a great strategy towards identifying news article bodies at scale.

**Additional Feedback:**

Expanding on data documentation and validation of data would be helpful to readers and users of the data.

**Clarity:**

The paper is overall well written. It missed discussions on limitations, and detailed discussions on quality control of text scraped and it’s alignment to headline associated with it in the data.

**Correctness:**

The dataset is large, has variety of topics represented and would be useful for the community for better language models.

**Documentation:**

Documentation for data is on the link provided, however the data card link within appears to be broken. In the link, sufficient documentation on collection, processing, and data fields are apparent. Intended usage of data, nad data validation documentation needs to be expanded on.

**Ethics:**

While authors say that the text is scraped from off-copyright newspapers, it is unclear if the copyright covered scraping for generative model training (since intended usage is not restricted to model testing alone).

I think a longer discussion on this is warranted or at least acknowledging this in limitations.

**Limitations:**

Authors have not discussed limitations of the methods used.
Would also recommend discussing the copyright issue, even though the newspapers are off-copyright. The point being, that curating articles at scale this way needs to be accompanied by a stronger statement about intended usage. For example, if models are to be trained based on this, does the copyright status still hold?

**Opportunities For Improvement:**

It isn’t clear if any human validation was done for data quality and if so how.

Expanding that section on data quality control would be useful for a user to know.

**Relation To Prior Work:**

Contributions are correctly claimed and situated within related work.

**Summary And Contributions:**

Authors contribute a large dataset composed of a variety of topics for the task of semantic similarity assessment of language models.
The use newspaper headlines and content for the same. Authors use bounding boxes to identify text body associated with headlines so as to be able to compose this dataset at scale.

---

> ### Author Response · Authors · 2023-08-22
> **Author Response**
>
> We thank the reviewer for their thoughtful comments. We direct all reviewers to our updated submission, and respond individually here to the points raised in this review.
>
> *“It isn’t clear if any human validation was done for data quality and if so how.”*
>
> Response: We carry out the following validation exercises (Section 4):
> 1. Quality of full article association, on an evaluation sample consisting of 214 scans
> 2. Quality of clustering to detect reproduced articles, on a hand-labeled test dataset consisting of 54,996 positive article pairs and over 100 million negative article pairs, created by labeling every front page article that appeared in our corpus of local newspapers during several  randomly selected days in the 1930s and 1970s
> 3. Quality of OCR, on a randomly selected set of 300 hand-annotated headlines per decade
>
> *“Authors have not discussed limitations of the methods used.”*
>
> Response: We have now added a section entitled “Limitations” (Section 6)
>
> *“Would also recommend discussing the copyright issue, even though the newspapers are off-copyright…”*
>
> Response: We now discuss this explicitly in Section 2. In the supplementary materials, we now include a table summarizing copyright law for works first published in the United States. The newspapers in the dataset are off-copyright, because they were published without a copyright notice or did not renew their copyright. The headlines were written by editors at these local papers, and since the papers either did not include a copyright notice or did not renew the copyright, these texts are unambiguously in the public domain. It is worth noting that copyright status is determined by copyright law at the date of publication of the headline. These laws were written many decades before generative AI was imaginable but are unambiguously clear that anyone can legally use or reference these works without permission.
>
> As we note in the paper, it is possible that a newspaper whose own content is in the public domain could reproduce copywritten content written by some third party. However, that is not a concern for this dataset. Headlines for wire articles were locally written – they are not syndicated content. If they had been syndicated, there would be no variation in the paired texts. If we were to accidentally include a syndicated headline, it would be dropped since we drop all pairs within a Levenshtein edit distance threshold. Also, an exhaustive search of U.S. copyright catalogs by [19] did not turn up a single instance of a wire service copyrighting the articles that these headlines describe. Even if they had, it would not pose a problem for this dataset, since it contains the headlines written by local newspapers.
>
> *Intended Usage and Validation*
>
> Response: We now include an explicit section on recommended usage (Section 6) and discuss validation in detail in Section 4.

---

### Official Review · Reviewer_d3Qx · 2023-07-24
**A good dataset: work in progress**

**Rating:** 5
**Confidence:** 5
**Correctness:** The claims made in the submission loo…

**Strengths:**

This size of the dataset is quite large compared to the existing datasets, and it is prepared on historical English.

**Additional Feedback:**

NA

**Clarity:**

The paper is well written. There are some typos, etc.
L131: Full stop is missing
Table 1 caption: This table ……. thefull → the full
At some places, the positive pairs have been mentioned as 400k. A mistake?

**Documentation:**

Yes, these details have been provided.

**Limitations:**

There is no discussion about the limitations of the paper.

**Opportunities For Improvement:**

While, overall, the dataset seems to be useful, sufficient details on the quality of the dataset are missing, for instance

How much is the dataset impacted by the OCR errors? While going through the Hugging Face repository, I see a lot of OCR errors in the initial years. Authors have not reported the error rates (across years), which may help understand how usable the pairs from initial years would be, without spelling correction. While it may not be possible to correct the entire dataset, is there a separate dev and test dataset that has been manually validated?

Additionally, what are the main tasks that can be defined over this dataset? The authors talk about abstractive similarity, and other datasets specific to entailment being complement to this. How challenging is this dataset? To understand that, a benchmarking would have been required, which is currently missing from this paper.

**Relation To Prior Work:**

The authors have clearly described relation to the prior work.

**Summary And Contributions:**

The paper presents a large semantic similarity dataset with ~400m parallel pairs using news headlines from historical US newspapers. A pipeline for performing OCR has been discussed, along with various heuristics used. The dataset is made available through Hugging Face.

---

> ### Author Response · Authors · 2023-08-22
> **Author response**
>
> We thank the reviewer for their thoughtful comments. We direct all reviewers to our updated submission, and respond individually here to the points raised in this review.
>
> *“What are the main tasks that can be defined over this dataset?"*
>
> Response: We now describe these tasks at the end of Section 2. HEADLINES is useful for training and evaluating models that aim to capture abstractive similarity, whether using the embeddings for tasks like clustering, nearest neighbor retrieval, or semantic search [18].. Because it contains chronological content over a long span of time, it can be used to evaluate dynamic language models for processing continuously evolving content [1, 15],, as well as how large language models can be adapted to process historical content [17, 5, 16]. Likewise, it can be used to train or evaluate models that predict the region or year a text was written [20]. In addition, it is useful for training models and developing benchmarks for a number of downstream tasks, such as topic classification of vast historical and archival documents, which have traditionally been classified by hand. This is an extremely labor intensive process, and as a result many historical archives and news collections remain largely unclassified. Similarly, it could facilitate creating a large scale dataset to measure semantic change, complementing existing small-scale SemEval tasks.
>
> *“How challenging is this dataset? To understand that, a benchmarking would have been required.”*
>
> Response: We have now added benchmarking (Section 5 and Figure 7), following the Massive Text Embedding Benchmark (MTEB), the primary benchmark in this domain. We use the MTEB clustering task, which embeds texts using different base language models and then uses $k$ - the number of clusters in the ground truth data - for k-means clustering. Following MTEB, we score the model using the v-measure (Rosenberg, 2007). In many real-world tasks, a problem is framed as a clustering task rather than as a classification task because $k$ is unknown. Nevertheless, we use this easier task where $k$ is assumed to be known for comparability to the literature.
>
> MTEB benchmarks clustering on Arxiv, Bioarxiv, Medarxiv, Reddit, StackExchange, and Twenty Newsgroups datasets. Texts are labeled with their classification (e.g., fields like ComputerVision for the Arxiv datasets; the subreddit for Reddit). The best average v-score across these datasets, from MPNet, is 43.69. The best average v-score across decades for HEADLINES, from ST5-XXL, is around 78 (Figure 7). This difference, while it could be interpreted as evidence that HEADLINES is easier, is also likely to reflect the fact that our cluster labels are less noisy, since texts in the same cluster summarize the same content. In contrast, titles of Reddit posts in the same subbredit may be only loosely (or not at all) linked to each other, and many could be within the domain of another subbredit cluster. While a user happened to post in one subreddit, another user could have reasonably made the same titled post in a different subreddit. The under-identification of the clustering tasks for some of the texts in MTEB is suggested by the very low v-scores across state-of-the-art language models. Overall, this suggests the high quality of HEADLINES relative to many web text datasets.  Yet there is still ample scope for improvement to the state-of-the-art model.
>
> *OCR errors*
>
> Response: Table 1 now includes an assessment of OCR errors, using three hundred randomly selected, hand annotated texts per decade. The character error rate ranges from 4.3% (1920s) to 1.5% (1980s), for the most part declining monotonically across time.. The character error rate in particular is higher for the 1920s, with the oldest (and hence noisiest) data happening to be the observations that display on huggingface since the dataset is sorted chronologically. The benchmarking exercise (Section 5) suggests that OCR errors are not overly-detrimental, and that the dataset is not overly noisy relative to existing datasets.
>
> *Typos*
>
> Response: We apologize for the typos and have carefully corrected them. The dataset size is almost 400 million.
>
> *Limitations*
>
> Response: Section 6 now discusses limitations.

---

### Official Review · Reviewer_boGz · 2023-07-26
**Interesting Dataset, some key creation details missing**

**Rating:** 5
**Confidence:** 4
**Clarity:** Yes. Above average for NeurIPS.

**Strengths:**

Overall, an interesting dataset that would be very useful for the field. Decent coverage of related work and fairly clear writing.

**Additional Feedback:**

I would really like to see this dataset published. It should be fairly minor changes in order to fully document everything.

**Correctness:**

I think so ... again, the lack of some key details make it hard for me to verify that it is constructed in a sound way.

**Documentation:**

Yes.

**Limitations:**

Yes

**Opportunities For Improvement:**

Explicitly detailing all of the steps of the dataset creation. For instance, the evaluation set ... or even just the size of the evaluation set. It would be hard to use the dataset and feel confident in it without some of these details.

**Relation To Prior Work:**

Yes. Nice related works section.

**Summary And Contributions:**

Overall, this is an interesting dataset with a lot of potential, but some key details are hard to follow from the writing of the paper. The paper looks at newspaper articles from a large time range and varying sources (within the US). Much of the paper is devoted to describing in a high-level how bounding boxes of text are associated with each other to combine to form an article. The details are in the supplementary materials (which feels like many of them should be in the main paper as they are integral to the dataset creation). Even within the supplementary material, I do not think that I would be able to reproduce their method easily. For instance, the “evaluation set” on page 6 in line 146 is never described. How large is it? Likewise, on page 8, line 173, there is a labeled set mentioned (but only at the decade level).

In general, this looks like a very interesting resource that I, as I’m sure many other people in the field, would be interested in using in their work. However, some key details about the dataset construction should be clarified to make this a more useful artifact for the community. In general, the writing is good (I would split section 4 into two subsections as it was a bit hard to see how reproduced content was different from bounding boxes) with a good coverage of related work and situating the dataset well within the community. With some minor, but very important revisions, to clarify some parts of the process, this would be an impactful dataset for the community.

---

> ### Author Response · Authors · 2023-08-22
> **Author response**
>
> We thank the reviewer for their thoughtful comments. We direct all reviewers to our updated submission, and respond individually here to the points raised in this review.
>
> *Lack of details about dataset construction and evaluation*
>
> Response: Section S1 in the supplementary now provides details on the evaluation set used for article association, and we have included some high level information about the evaluation set in the main text. The set used to evaluate performance on detecting duplicated content is now also described in detail in the  supplementary material and additional details about this evaluation have been added to the main text. We have also added an evaluation of the OCR quality in Table 1. We have aimed to fit as many details as possible into the main text - in Section 4- given the space constraints. The supplementary materials have also been significantly expanded, to provide all details required to reproduce the dataset. Finally, a benchmarking exercise (Section 5) suggests the high quality of the dataset.
>
> *"I would split section 4 into two subsections as it was a bit hard to see how reproduced content was different from bounding boxes with a good coverage of related work and situating the dataset well within the community."*
>
> Response: We agree and have split Section 4 accordingly.

---

### Official Review · Reviewer_ZfNZ · 2023-08-01
**A Massive Scale Semantic Similarity Dataset of Historical English**

**Rating:** 5
**Confidence:** 5
**Correctness:** yes
**Clarity:** yes, but not a well-written paper in …

**Strengths:**

Data Source and Coverage: Utilizing newly digitized articles from off-copyright, local U.S. newspapers spanning nearly 70 years (1920-1989) is a significant strength. The dataset covers a large time span, providing researchers with the opportunity to study semantic change over time and understand historical linguistic patterns.

Data Quality: The authors have taken care to address potential noise and abridgement in the dataset due to OCR and article association errors. The rule-based approach with high precision and the use of RoBERTa bi-encoder for detecting reproduced content demonstrate a thoughtful and rigorous data processing methodology.

Potential for Research: The paper discusses how the dataset can be used for a variety of tasks, including contrastively trained semantic similarity models, geographic analysis based on authors' locations, and studying semantic change across space and time. This highlights the dataset's potential for diverse and meaningful research applications.

**Additional Feedback:**

The paper is a good contribution but the writing style of the paper is more suitable for art/history conferences not CS. To improve the work, the authors could add more statistical results and evaluation, tables and ML visualization.

**Documentation:**

yes

**Opportunities For Improvement:**

Noisy OCR and Article Association: The dataset's construction relies on OCR and rule-based article association, which introduces potential noise and errors. Although the authors address this by using a rule-based approach with high precision, there might still be cases where articles are incorrectly associated or abridged. This noise may impact the quality of the dataset and the performance of models trained on it.

Historical Bias: The dataset covers newspaper articles from local U.S. newspapers spanning nearly 70 years, which introduces a historical bias. The linguistic patterns, writing styles, and vocabulary used in these articles may differ significantly from contemporary language. Researchers need to be cautious when using this dataset for tasks that require current language understanding and generalization.

Lack of Diversity in Topics: As the dataset is derived from newspaper articles, the topics covered may be biased towards news events and local issues prevalent in the U.S. While this is valuable for certain historical analyses, it may not be representative of the diverse range of topics seen in other datasets or real-world language use.

**Relation To Prior Work:**

no

**Summary And Contributions:**

The study introduces the HEADLINES dataset, a massive-scale semantic similarity dataset created from digitized articles in off-copyright, local U.S. newspapers spanning nearly 70 years (1920-1989). The dataset contains nearly 400,000 positive semantic similarity pairs, where the headlines serve as abstractive summaries of the associated articles. The uniqueness of this dataset lies in its historical context, covering a long time span and providing valuable information about geographic locations of the authors. Compared to existing semantic similarity datasets, HEADLINES significantly exceeds their size and captures a different type of semantic similarity. The authors describe the dataset construction process, which involves associating headlines with their corresponding articles using a combination of layout information and language understanding. They also discuss the methodology for detecting reproduced content in the articles. The HEADLINES dataset opens up new possibilities for contrastively trained semantic similarity models, enabling research on semantic change over time and various other applications. The paper provides a comprehensive overview of the dataset's construction and its potential significance in the field of semantic similarity modeling. The researchers demonstrate that they have considered historical context, coverage, and data quality while constructing the dataset.

---

> ### Author Response · Authors · 2023-08-22
> **Author response**
>
> We thank the reviewer for their thoughtful comments. We direct all reviewers to our updated submission, and respond individually here to the points raised in this review.
>
> *“Noisy OCR and Article Association: The dataset's construction relies on OCR and rule-based article association, which introduces potential noise and errors.”*
>
> Response: The F1 for article association is 93.7. Importantly, we use the rule-based approach only for associating articles where precision is nearly perfect, and associate remaining cases using a RoBERTa Base cross-encoder model. Indeed, if we had used rules alone, there would have been far more noise.  Table 1 now includes an assessment of OCR errors, using three hundred randomly selected, hand annotated texts per decade. The character error rate ranges from 4.3% (1920s) to 1.5% (1980s), for the most part declining monotonically across time.
>
> We do not think OCR errors are overly-detrimental to dataset quality. We now conduct a benchmarking exercise (Section 5), using a variety of different language models and a clustering evaluation task from the Massive Text Embedding Benchmark (MTEB). The MTEB clustering task embeds texts using different base language models and then uses $k$ - the number of clusters in the ground truth data - for k-means clustering. Following MTEB, we score the model using the v-measure (Rosenberg, 2007). The best average v-score across decades for HEADLINES, from ST5-XXL, is around 78 (Figure 7), significantly higher than the corresponding score on the average MTEB task. Overall, this suggests the high quality of HEADLINES, yet there is still ample scope for improvement to the state-of-the-art model.
>
> *The dataset contains historical biases*
>
> Response: We completely agree, though we see this as a feature rather than a bug. There are many dozens of datasets and models trained on modern web texts for tasks requiring current language understanding and generalization, but very little data for tasks that require historical language understanding or seek to study how language has changed across time. There is considerable interest in tasks that require historical language understanding or world knowledge, but few large-scale datasets to serve this community. In the limitations section, we now explicitly recommend against using HEADLINES for tasks that require texts that fully conform to current cultural standards or semantic norms.
>
> *Lack of diversity in topics*
>
> We understand this concern but would also like to underscore the remarkable breadth of topics that enters the nearly 400 million headline pairs in the dataset. Historically, local newspapers served a purpose akin to the internet today, providing people with a wealth of information about different topics. In addition to domestic and international news stories, there is a diversity of content about topics including e.g., cultural and entertainment; cooking, cleaning, and beauty advice; sports; religion; scientific advances; health; pets; and a vast diversity of other topics.
>
> If HEADLINES were highly specialized, it would tend to be quite different across the board from the many datasets included in the Massive Text Embedding Benchmark (MTEB). However, we now perform a quantitative exercise, calculating the similarity between HEADLINES and the varied MTEB benchmarks, that shows that this is not the case (Figure 4). This exercise is described in detail in Section 3 on related literature. This exercise also provides quantitative evidence that HEADLINES is if anything more diverse than the typical MTEB benchmark (see the discussion in lines 123-140).
>
> We agree that headlines are a specific linguistic form (that changes over time). However, this is a feature that most of the datasets in this domain share (e.g., tweets, question-answer pairs, etc), and an important reason why the community can benefit from a diversity of datasets.
>
> *Add more statistical results and evaluation, tables and ML visualization.*
>
> We agree with this and have added several analyses. We measure the similarity of HEADLINES across years (Section 2 and Figure 3), the similarity of HEADLINES to other texts in the Massive Texts Embedding Benchmark (MTEB) (Section 3 and Figure 4), and benchmark HEADLINES following the standards of MTEB, the largest benchmark in this domain (Section 5).

---

### Decision · Program_Chairs · 2023-09-22

**Decision:**

Accept (Poster)

**Comment:**

The paper presents HEADLINES a very large diachronic dataset including nearly 400,000 semantic similarity documents pairs. I don’t think the limitations identified by the reviewers would justify the rejection of this paper, which can be an interesting contribution to NeurIPS and to the NLP community. The authors provided a thorough reply to the reviewers and will surely be able to improve the final version of the paper.